# Scaling Symbolic Methods using Gradients for Neural Model Explanation

**Subham S. Sahoo, Subhashini Venugopalan, Li Li, Rishabh Singh & Patrick Riley**
Google Research
{subhamsahoo,vsubhashini,leeley,rising,pfr}@google.com

## Abstract

Symbolic techniques based on Satisfiability Modulo Theory (SMT) solvers have been proposed for analyzing and verifying neural network properties, but their usage has been fairly limited owing to their poor scalability with larger networks. In this work, we propose a technique for combining gradient-based methods with symbolic techniques to scale such analyses and demonstrate its application for model explanation. In particular, we apply this technique to identify minimal regions in an input that are most relevant for a neural network's prediction. Our approach uses gradient information (based on Integrated Gradients) to focus on a subset of neurons in the first layer, which allows our technique to scale to large networks. The corresponding SMT constraints encode the minimal input mask discovery problem such that after masking the input, the activations of the selected neurons are still above a threshold. After solving for the minimal masks, our approach scores the mask regions to generate a relative ordering of the features within the mask. This produces a saliency map which explains "where a model is looking" when making a prediction. We evaluate our technique on three datasets - MNIST, ImageNet, and Beer Reviews, and demonstrate both quantitatively and qualitatively that the regions generated by our approach are sparser and achieve higher saliency scores compared to the gradient-based methods alone. Code and examples are at -
https://github.com/google-research/google-research/tree/master/smug_saliency

## 1 Introduction

Satisfiability Modulo Theory (SMT) solvers (Barrett & Tinelli, 2018) are routinely used for symbolic modeling and verifying correctness of software programs (Srivastava et al., 2009), and more recently they have also been used for verifying properties of deep neural networks (Katz et al., 2017). SMT Solvers in their current form are difficult to scale to large networks. Model explanation is one such domain where SMT solvers have been used but they are limited to very small sized networks (Gopinath et al., 2019; Ignatiev et al., 2019). The goal of our work is to address the issue of scalability of SMT solvers by using gradient information, thus enabling their use for different applications. In this work, we present a new application of SMT solvers for explaining neural network decisions.

Model explanation can be viewed as identifying a minimal set of features in a given input that is critical to a model's prediction (Carter et al., 2018; Macdonald et al., 2019). Such a problem formulation for identifying a minimal set lends itself to the use of SMT solvers for this task. We can encode a neural network using real arithmetic (Katz et al., 2017) and use an SMT solver to optimize over the constraints to identify a minimal set of inputs that can explain the prediction. However, there are two key challenges in this approach. First, we cannot generate reliable explanations based on final model prediction as the minimal input is typically out of distribution. Second, solving such a formulation is challenging for SMT solvers as the decision procedures for solving these constraints have exponential complexity, and is further exacerbated by the large number of parameters in typical neural network models. Thus, previous approaches for SMT-based analysis of neural networks have been quite limited, and have only been able to scale to networks with few thousands of parameters.

To solve these challenges, instead of doing minimization by encoding the entire network, our approach takes advantage of the gradient information, specifically Integrated Gradients (IG) (Sundararajan et al., 2017), in lieu of encoding the deeper layers, and encodes a much simpler set of linear constraints

pertaining to the layer closest to the input. We encode the mathematical equations of a neural network as SMT constraints using the theory of Linear Real Arithmetic (LRA), and use z3 solver (Bjørner et al., 2015) as it additionally supports optimization constraints such as minimization. The SMT solver then finds a minimal subset (of input features) by performing minimization on these equations. Thus, our approach, which we refer to as SMUG, is able to scale **S**ymbolic **M**ethods **U**sing **G**radient information while still providing a faithful explanation of the neural network's decision.

SMUG is built on two properties. First, based on the target prediction, SMUG uses gradient information propagated from the deeper layers to identify neurons that are important in the first layer, and only encodes those. For this, we use IG (Sundararajan et al., 2017) instead of relying on gradients alone. Second, for the optimization, a set of input pixels are determined to be relevant for prediction if they are able to activate the neurons deemed important, and maintain a fraction of their activation as the original (full) input image. Empirically, we observe good performance on visual and text classification tasks.

We evaluate SMUG on three datasets: MNIST (LeCun et al., 2010), ImageNet (Deng et al., 2009), and Beer Reviews (McAuley et al., 2012). We show that we can fully encode the minimal feature identification problem for a small feedforward network (without gradient-based neuron selection) for MNIST, but this full SMT encoding scales poorly for even intermediate sized networks. On ImageNet, we observe that our method performs better than Integrated Gradients (Sundararajan et al., 2017) and several strong baselines. Additionally, we observe that our approach finds significantly sparser masks (on average 17% of the original image size). Finally, we also show that our technique is applicable to text models where it performs competitively with other methods including SIS (Carter et al., 2018) and Integrated Gradients (Sundararajan et al., 2017).

This paper makes the following key contributions:

- We present a technique (SMUG) to encode the minimal input feature discovery problem for neural model explanation using SMT solvers. Our approach, which does masking on linear equations also overcomes the issue of handling out-of-distribution samples.

- Our approach uses gradient information to scale SMT-based analysis of neural networks to larger models and input features. Further, it also overcomes the issue of choosing a "baseline" parameter for Integrated Gradients (Kapishnikov et al., 2019; Sturmfels et al., 2020).

- We empirically evaluate SMUG on image and text datasets, and show that the minimal features identified by it are both quantitatively and qualitatively better than several baselines.

- To improve our understanding on saliency map evaluation, we show how the popular and widely used LSC metric (Dabkowski & Gal, 2017) can be gamed heuristically to generate explanations that are not necessarily faithful to the model (Sec. 5.2)

## 2 RELATED WORK

SMT based symbolic techniques have been used for verifying neural network properties (Huang et al., 2017; Katz et al., 2017). Reluplex (Katz et al., 2017) extends the simplex method to handle ReLU functions by leveraging its piecewise linear property and presents an iterative procedure for gradual satisfaction of the constraints. (Huang et al., 2017) proposes a layer-wise analysis using a refinement-based approach with SMT solvers for verifying the absence of adversarial input perturbations. (Zhang et al., 2018) present a linear programming (LP) formulation again using the piecewise linear property of ReLU to find minimal changes to an input to change a network's classification decisions. (Gopinath et al., 2019) uses Reluplex to learn input properties in the form of convex predicates over neuron activations, which in turn capture different behaviors of a neural network. While SMT based techniques from (Gopinath et al., 2019), (Ignatiev et al., 2019) have shown promising results, they only scale to Neural Network with 5000 nodes. Thus, scaling these approaches for larger neural networks and performing richer analysis based on global input features still remains a challenge. We present and demonstrate an approach that works for larger and more complex image and text models.

While most of the above SMT based techniques focus on verifying properties of deep networks, our work focuses on applying symbolic techniques to the related task of model explanation, i.e. to say where a model is "looking", by solving for the input features responsible for a model's prediction.

Some explanation techniques are model agnostic (i.e., black-box) while others are back-propagation based. Model agnostic (black-box) explanation techniques such as SIS, LIME (Alvarez-Melis & Jaakkola, 2018; Carter et al., 2018) have a similar formulation of the problem as ours in the sense that they perturb the input pixels by masking them and optimize to identify minimal regions affecting the performance of the model. This formulation can lead to evaluating the model on out of distribution samples (Hooker et al., 2019) with potential for adversarial attacks (Slack et al., 2020). In contrast, back-propagation based methods (Bach et al., 2015; Sundararajan et al., 2017; Selvaraju et al., 2016) examine the gradients of the model with respect to an input instance to determine pixel attribution. Our work builds on the IG method (Sundararajan et al., 2017). IG integrates gradients along the "intensity" path where the input (image or text embedding) is scaled from an information-less baseline (all zeros input, e.g., all black or random noise image) to a specific instance. This helps the model determine attribution at the pixel level. In our work, we use IG to determine important nodes in the first layer (closest to the input). The key improvement of our technique over IG is that, by using IG only on the first layer and then using SMT solver based solution to determine saliency on the input, we not only preserve faithfulness, but also overcome the issue of choosing an appropriate baseline image for IG (Kapishnikov et al., 2019; Sturmfels et al., 2020; Xu et al., 2020).

# 3 METHOD: SCALING SYMBOLIC METHODS USING GRADIENTS (SMUG)

We describe our approach SMUG, which combines attribution based on gradient information with an SMT-based encoding of the minimal input feature identification problem. We then show how to generate saliency maps from the predicted boolean mask for image and text applications.

## 3.1 CHALLENGES WITH ENCODING NEURAL NETWORKS FOR EXPLANATION

Previous works (Katz et al., 2017) have shown it's possible to use SMT solvers to encode the semantics of neural networks with specific activation functions, while others (Carter et al., 2018; Macdonald et al., 2019) have used the idea of minimality to identify importance of inputs in the context of model explanation. In particular, given a neural network $N_\theta$ ($\theta$ denotes the parameters), let $X \in \mathbb{R}^{m \times n}$ denote an input image with $m \times n$ pixels, $M \in \{0, 1\}^{m \times n}$ an unknown binary mask, and $L_i(\cdot)$ the output (i.e., activations) of the $i^{th}$ layer. Also, let $L_{|L|-1}(N_\theta(X))$ denote the logits of the final layer. Previous works (Carter et al., 2018; Macdonald et al., 2019) have used the following formulation for explanation.

$$\min(\sum_{ij} M_{ij}) : \operatorname{argmax} L_{|L|-1}(N_\theta(X)) = \operatorname{argmax} L_{|L|-1}(N_\theta(M \odot X))$$

However, there are two key problems with this formulation. First, the masked inputs found by solving this formulation are typically out of training distribution, so explanation techniques based on model confidence over a forward pass of the masked inputs (such as the above formulation) no longer produce reliable explanations. Second, using such a formulation directly with SMT solvers is not tractable as the number of constraints in the encoding grows linearly with increasing network size and SMT decision procedure for Non-linear Real Arithmetic is doubly exponential. Thus, when we apply such an encoding even for a small feed-forward network on MNIST dataset, the SMT solver does not scale well (Section 4, Appendix A).

## 3.2 OUR SOLUTION: COMBINING GRADIENTS WITH SYMBOLIC ENCODING

Our proposal is to use gradient based attributions to overcome both these issues: 1) as a proxy for the deeper layers (for scalability), and 2) to identify a subset of neurons in the first layer capturing information crucial for the model's prediction, and encode them using SMT which avoids performing a forward pass on out-of-distribution inputs.

Specifically, we use Integrated Gradients (IG) (Sundararajan et al., 2017) to score the neurons in order of relevance by treating the first layer activations ($L_1$) as an input to the subsequent network. This assigns an attribution score to each hidden node where a node with a positive score means that the information captured by that neuron is relevant to the prediction and a node with a negative / zero score is considered irrelevant. More specifically, suppose $F : R^{a \times b \times c} \to [0, 1]^d$ represents a deep network and $x$ represents the input. F often refers to the full network (from the input to the

final softmax) but in our case, it refers to the nodes in the network second through the final layer. a, b, and c refer to the dimensions of the tensor input to the network (analogous to width, height, and channels) and d is the number of output nodes in the final softmax layer (number of classes). Integrated gradients are obtained by accumulating the gradients at all points along the straight line path from an "information-less" baseline $x'$ to the input $x$. The information-less baseline in this case is an all zeros tensor. The path can be parameterized as $g(\alpha) = x' + \alpha \cdot (x - x')$. IG is then given by Eq. 1.

$$IG(x) = (x - x') \int_{\alpha=0}^{1} \frac{\partial F(g(\alpha))}{\partial g(\alpha)} d\alpha \qquad (1)$$

**Top-k Neurons.** The second key idea of our approach is that we only consider activations with the highest positive attributions. Empirically, we still observe scaling issues with SMT when considering all first layer neurons ($L_1$) with +ve attributions, so a method for picking a subset of neurons (top-k attributions) is important for practical application.

With these two ideas, we can now formulate our problem as follows. Let, $D^k$ represent the set of k important nodes with the highest attributions in $IG(N_\theta(X))$, and $\gamma$ be a parameter which regulates how "active" the neurons remain after masking[1].The goal now is to learn a mask $M$ such that:

$$\min(\sum_{ij} M_{ij}) : L_1(N_\theta(M \odot X))_t > \gamma \cdot L_1(N_\theta(X))_t \qquad \forall t \in D^k \qquad (2)$$

**Inducing Sparsity**. IG ensures that the information captured by the neurons in $D^k$ are relevant for the model's prediction. Thus, the set of features corresponding to these neurons contributes towards the relevant information. Let's denote this set as $S$. To filter less relevant features from $S$ we introduce a notion of minimization. By inducing sparsity we wish to remove any false positives from $S$. Thus, we define "relevant features" as a minimal subset of features in $S$ that causes the nodes in $D^k$ to remain active. The notion being - if for a masked input the "important nodes" (w.r.t the original input) are active then the information relevant for the prediction of the original image is contained in the mask. Thus, the mask highlights features relevant for the model's prediction.

Thus, this formalism allows us to come up with constraints which don't involve a full forward pass of the masked image while at the same time keeping things scalable for the SMT solver.

## 3.3 SMT FORMULATION OF MINIMAL INPUT MASK DISCOVERY PROBLEM

Given $D^k$, a set of $k$ neurons with highest positive attributions in $IG(N_\theta(X))$, our goal is to find a minimal mask such that the activations of these neurons are above some threshold ($\gamma$) times their original activation values. Eq. 3 shows the constraints for a minimal mask. The first set of constraint specifies that the unknown mask variable $M$ can only have 0 and 1 as possible entries in the matrix. The second and third set of constraints encode the activation values of the first layer of network with corresponding masked and original inputs respectively. The fourth set of constraint states that the activations of these $k$ neurons should be at least $\gamma$ times the original activation values, and the final constraint adds the optimization constraint to minimize the sum of all the mask bits. Note that here we show the formulation for a feedforward network and a input $X$ with 2 channels, but it can be extended to convolutional networks and an input with $3^{rd}$ channel as well in a straightforward manner where the mask variables across the same channel share the same mask variable.

$$\exists M : \bigwedge_{1 \leq i \leq m, 1 \leq j \leq n} (M_{ij} == 0) \vee (M_{ij} == 1) \bigwedge_{\forall i \in D^k} o_i^m = (W_1(X \odot M) + b_1)_i$$

$$\bigwedge_{\forall i \in D^k} o_i = (W_1 X + b_1)_i \bigwedge_{\forall i \in D^k} o_i^m > \gamma \cdot o_i \bigwedge \mathbf{minimize}(\Sigma_{ij} M_{ij}) \qquad (3)$$

## 3.4 CONSTRUCTING SALIENCY MAP FROM BINARY MASK

The SMT solver generates a minimal binary input mask by solving the constraints shown in Eq. 3. We further use the IG attribution scores for the hidden nodes in the first layer to assign importance

---

[1]This formulation is valid for networks with ReLU activations and would need to be modified for a different choice of activation such as tanh.

scores to mask pixels associated with that hidden node. A mask variable $M_{ij}$ that is assigned a value of 1 by SMT is assigned a score $s_{ij}$ computed as:

$$s_{ij} = \sum_{1 \leq p \leq k} \alpha(o_p)\mathbb{1}_{\texttt{receptive}(o_p)}(x_{ij}) \qquad \forall i,j : M_{ij} = 1 \qquad (4)$$

where $\alpha(o_p)$ denotes the attribution score assigned by IG for neuron $o_p$ and the indicator function denotes that pixel $x_{ij}$ is present in the receptive field of $o_p$, i.e. it is present in the linear SMT equation used to compute $o_n$. These scores are then used to compute a continuous saliency map for an input (Appendix Fig. 9). Finally, to amplify the pixel differences for visualization purposes in gray scale, we scale the non-zero score values between 0.5 and 1.

### 3.5 Handling out-of-distribution masked inputs

Several methods - (Alvarez-Melis & Jaakkola, 2018), (Petsiuk et al., 2018), (Fong & Vedaldi, 2017) optimize their masks by doing a forward pass on perturbed images that are potentially out-of-distribution. However, SMUG performs forward pass of the masked image only till the first layer (instead of the final layer) to generate linear equations as described in Sec. 3.3. Since, the constraints are linear 6, minimization can be thought of as analogous to examining coefficients for a linear regression model (in linear regression important attributes are the ones with the largest coefficients). So, in our case we do indeed mitigate the out-of-distribution issue as the full forward pass of the masked input isn't present.

## 4 Experimental Setup

**Datasets.** We empirically evaluate SMUG on two image datasets, MNIST (LeCun et al., 2010), and ImageNet (Deng et al., 2009), as well as a text dataset of Beer Reviews from (McAuley et al., 2012).

**MNIST:** We use the MNIST dataset to show the scalability of the full network encoding in SMT (presented in Section 3.1). We use a feedforward model consisting of one hidden layer with 32 nodes (ReLU activation) and 10 output nodes with sigmoid, one each for 10 digits (0 - 9). For 100 images chosen randomly from the validation set, the SMT solver could solve the constraint shown in Eq. 6 (returns SAT) for only 41 of the images. For the remaining 59 images, the solver returns UNKNOWN, which means the given set of constraints was too difficult for the solver to solve.

**ImageNet:** We use 3304 images ($224 \times 224$) with ground truth bounding boxes from the validation set of ImageNet. The images for which the model classification was correct and a ground truth bounding box annotation for the object class was available were chosen. We use the Inception-v1 model from (Szegedy et al., 2015) which classifies images into one of the 1000 ImageNet classes.

**Beer Reviews:** To evaluate SMUG on a textual task we consider the review rating prediction task on the Beer Reviews dataset[2] consisting of 70k training examples, 3k validation and 7k test examples. Additionally, the dataset comes with ground truth annotations where humans provide the rationale (select words) that correspond to the rating and review. We train a 1D CNN model to predict the rating for the *aroma* of the beer on a scale from 0 to 1. This model is identical to the model used in (Carter et al., 2018) and consists of a convolution layer with 128 kernels followed by a ReLU, a fully connected layer, and a sigmoid. It achieves a validation mean square error of 0.032.

**Metrics.** Assessing the quality of the saliency maps, especially binary masks, is challenging. The change in confidence of the classifier (between the original and masked) image alone may not be a reliable measure since the masked input could fall out of the training distribution (Hooker et al., 2019). Instead, we use the metric proposed in (Dabkowski & Gal, 2017) shown in Eq. 5. This metric, which we term Log Sparsity Confidence difference (**LSC**) score, first finds the tightest bounding box that captures the entire mask, then computes confidence on the cropped box resized to the original image size (we use bilinear interpolation). This not only helps keep images closer to the training distribution, but also helps evaluate explanations without the need for groundtruth annotations. The LSC score is computed as:

$$LSC(a,c) = \log(\tilde{a}) - \log(c), \qquad \tilde{a} = \max(0.05, a) \qquad (5)$$

---

[2]http://people.csail.mit.edu/taolei/beer/

where $a$ is the fractional area of the rectangular cropped image and $c$ is the confidence of the classifier for the true label on the cropped image. A saliency map that is compact and allows the model to still recognize the object would result in a lower LSC score. LSC captures model confidence as well as compactness of the identified salient regions, both of which are desirable when evaluating an explanation. The compactness in particular also makes it suitable for evaluating SMT based methods for the effect of minimization. We adapt LSC to also assess continuous valued saliency maps by, 1) setting a threshold on the continuous valued saliency map to convert them to a binary mask and 2) iterating over multiple thresholds (in steps) to identify the one that results in the best LSC score. We also report the fraction of images for which the mask generated by a given method is better (i.e. produces an equal or lower LSC score) than other methods, which we refer to as **Win%**.

**Comparison methods.** The final saliency mask for SMUG comes from Eq. 4 (Sec. 3.4). We compare this to the saliency maps, and bounding boxes from several baselines described below.
**SMUGbase** is a variant of SMUG that does not perform SMT-based minimization. Here, in Eq. 2, we simply set $M_{ij} = 1$ for each pixel $x_{ij}$ that is in the receptive fields of the top-$k$ neurons in the first layer ($L_1$) selected by IG. We note here that in case of both SMUG and SMUGbase in the formulations in Eq. 2 and 3, we set $k = 3000$, $\gamma = 0$ for ImageNet, and $k = 100$, $\gamma = 0$ for text experiments (this choice is explored more in the supplementary material).
**IG** corresponds to Integrated Gradients (Sundararajan et al., 2017) with the black image as a baseline.
**GRADCAM** (Selvaraju et al., 2017) uses a weighted average of the CNN filters for saliency, with weights informed by the gradients.
**SIS** refers to Sufficient Input Subset (Carter et al., 2018), which finds multiple disjoint subsets of input features (in decreasing order of relevance) which individually allow a confident classification. SIS did not scale for ImageNet and we only compare against it on the text dataset.
**GROUNDTRUTH** corresponds to the baseline that uses human annotated bounding box, which capture the object corresponding to the image label.
**MAXBOX** denotes maximal mask spanning the entire image.
**CENTERBOX** uses a bounding box placed at the center of the image covering half of the image area.
**OPTBOX** refers to a bounding box that approximately optimizes for LSC. The saliency metric in Eq. 5 relies on finding a single bounding box for an image. To find a box that directly maximizes the metric, we first discretize the image into subgrids of $10 \times 10$ pixels; and then perform a brute force search by selecting 2 points on the grid (to represent opposite corners of a rectangle) and identify a subgrid with the best score.

## 5 RESULTS

### 5.1 IMAGENET

As mentioned previously, when computing masks using SMUG, for ImageNet we set $k = 3000$ in Eq. 2 and 3. Further, each masking variable $M_{ij}$ is used to represent a $4 \times 4$ grid of pixels instead of a single pixel (to reduce running time). Table 1 presents quantitative results reporting the median LSC score and Win% values. Fig. 1 present qualitative examples

Table 1: **ImageNet.** We report the median LSC score ($\downarrow$ lower is better) along with the 75th (top) and 25th percentile (bottom) values, and the mean Win% score in percentage ($\uparrow$ higher is better) with binomial proportion confidence interval (normal approximation) on 3304 images in the validation set. The Win% values don't sum to 100 due to overlap when methods achieve identical scores. SMUG and SMUGbase receive similar LSC scores which suggests that the SMT based minimization retained the relevant regions while successfully removing about 66% pixels from SMUGbase. We also report the average size of the masks as a fraction of the total image area (sparsity) with normal confidence interval for 3304 images. Sparsity scores for IG and Gradcam are omitted because they aren't masking methods.

| Method | SMUG | SMUGbase | GROUNDTRUTH | IG | GRADCAM | CENTERBOX | MAXBOX | OPTBOX |
|---|---|---|---|---|---|---|---|---|
| **LSC** $\downarrow$ | $\mathbf{-1.26}^{-0.75}_{-1.80}$ | $-1.23^{-0.71}_{-1.76}$ | $-0.34^{0.04}_{-0.81}$ | $-0.29^{-0.05}_{-0.62}$ | $-1.10^{-0.50}_{-1.67}$ | $-0.64^{-0.29}_{-0.69}$ | $0.04^{0.23}_{0.00}$ | $-2.27^{-1.79}_{-2.71}$ |
| **Win%** $\uparrow$ | $\mathbf{40.9} \pm 1.68$ | $33.5 \pm 1.61$ | $3.6 \pm 0.64$ | $1.7 \pm 0.44$ | $37.8 \pm 1.65$ | $2.6 \pm 0.54$ | $0.2 \pm 0.16$ | - |
| **Sparsity%** $\downarrow$ | $\mathbf{17.7} \pm 0.10$ | $43.3 \pm 0.32$ | $50.7 \pm 0.98$ | - | - | $50.0 \pm 0.99$ | $100.0 \pm 0.0$ | $8.9 \pm 0.0$ |

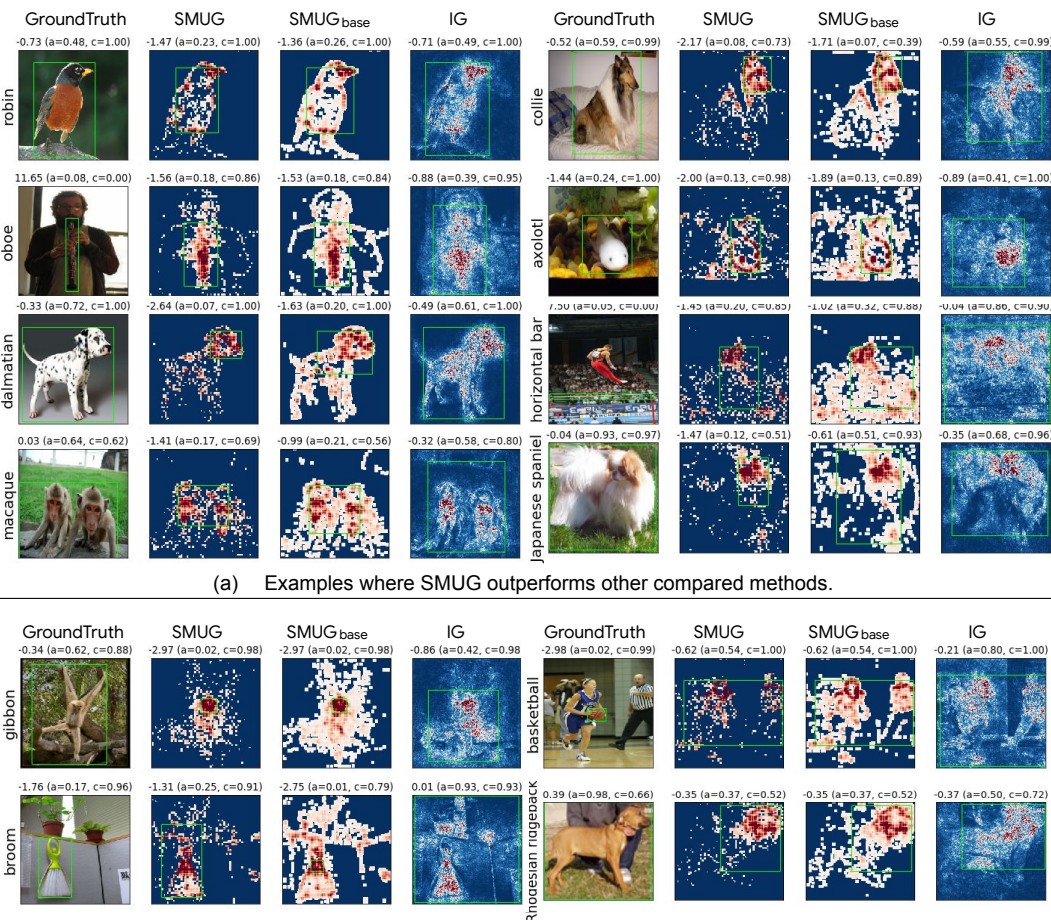

Figure 1: Examples where SMUG outperforms other compared methods (top 4 rows), and where SMUG performs less favorably (last 2 rows) based on LSC. The green box on the original image highlights the groundtruth box; for the saliency methods it represents the bounding box with the best LSC score. Numbers on top denote the LSC score, the fractional area of the bounding box ($a$), and the confidence of the classifier ($c$) on the cropped region. Red and blue colors denote regions of high and low importance respectively. More qualitative examples can be found in Appendix D

**IG vs SMUG and SMUG_base.** From Table 1, we observe that SMUG and SMUG$_{\text{base}}$ achieve a significantly better score ($-1.26$ and $-1.23$ resp.) compared to IG ($-0.34$). As observed in some qualitative examples (Fig. 1, Appendix D), SMUG tends to assign high scores to a much more localized set of pixels whereas IG distributes high scores more widely (spatially). As LSC metric favors compactness, which is desirable for human interpretability, it results in better scores for SMUG and SMUG$_{\text{base}}$.

**Choice of baseline for IG.** Another reason why SMUG$_{\text{base}}$ and SMUG outperform IG is that, they apply IG to the first layer of the network (as opposed to the input/image layer). IG attribution in the pixel space is known to be noisy (Smilkov et al., 2017), further attributions produced by IG depend on the choice of the baseline (Kapishnikov et al., 2019; Xu et al., 2020; Sturmfels et al., 2020). The reason for this can be observed from Eq. 1. In Eq. 1, $x'$ represents the baseline "information-less" image. Based on this, the input dimensions close to the baseline receive very low attributions even though they might be important. i.e., if $i, j$ denote pixel locations, when $x_{i,j} - x'_{i,j} \approx 0$, the attribution $IG_{i,j}(x) \approx 0$ irrespective of how important the pixels are. For instance, black pixels (RGB value of $(0, 0, 0)$) will receive an exact 0 attribution for a black baseline. In fact, for any baseline, IG will be insensitive to the dimensions close to the baseline value. When IG is applied to the first layer activations however, the nodes with near 0 activations will by default receive less attribution. Thus, we believe that **0** activations in the first layer is a more natural baseline for IG for ReLU networks, which is quantitatively observable in better LSC scores.

**SMUG vs SMUG$_{base}$.**

Based on the LSC scores in Table 1, SMUG narrowly outperforms SMUG$_{base}$. Recall however, that SMUG is a sparser version of SMUG$_{base}$ obtained from the minimization constraints of the SMT solver (Eqns. 2 and 3). LSC scores use the sparsity of the bounding boxes as a proxy for the sparsity of the saliency map. As we can see in Fig. 1(a) and 1(b) that the bounding boxes for SMUG and SMUG$_{base}$ have similar sizes even though the former produces much sparser saliency maps. This causes both SMUG and SMUG$_{base}$ to receive similar LSC scores. Thus, the LSC scores alone don't reveal the full picture and we need to consider the **sparsity** as well. This is defined as the fraction of the pixels with non-zero attributions to the total number of pixels in the image. We find that the average SMUG$_{base}$ mask has a sparsity of **43%** while the average SMUG mask has a sparsity of just **17%**. This is also evident from the examples in Fig. 1. SMUG and SMUG$_{base}$ receive similar LSC scores which suggests that the symbolic encoding successfully retains pixels relevant to the prediction while removing about 66% pixels from SMUG$_{base}$.

**GROUNDTRUTH, CENTERBOX.** Based on qualitative examples Figs. 1, we can observe that in almost all cases the object is at the center of the image. Hence, CENTERBOX is likely to capture some part of the image. Further, a fair number of objects are large covering much of the image e.g., Fig. 1 *Macaque, Collie, Robin, Dalmation*. In these cases, the groundtruth bounding boxes are also large to fully cover all pixels corresponding to the object. In contrast, SMUG saliency maps are more compact for both large and small objects, and hence achieve a better LSC score.

## 5.2 DISCUSSION: ANALYZING THE LSC METRIC AND OPTBOX

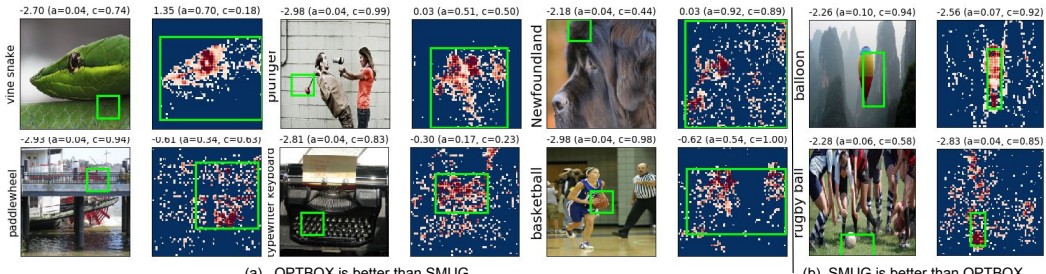

Figure 2: OPTBOX vs SMUG. 6 columns on the left correspond to the images for which OPTBOX gets a better score than SMUG. 2 columns on the right correspond to the images for which SMUG got a better LSC score than OPTBOX. Numbers at the top denote the LSC score, fractional area of the bounding box $a$ and the confidence of the classifier $c$ on the cropped region.

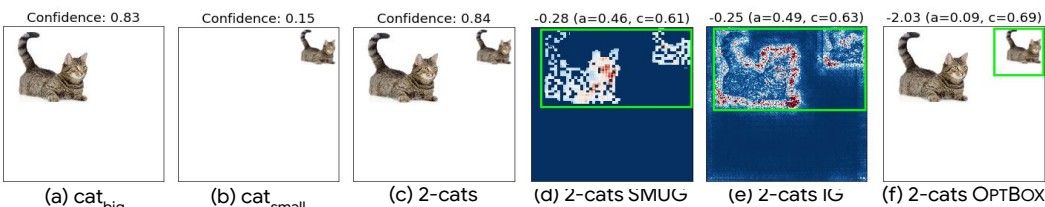

Figure 3: (a) shows an image of a cat (cat$_{big}$) placed on a white background that is classified with a confidence of 0.83. (b) shows an image of the same cat (cat$_{small}$), scaled to a quarter of its original size, that is classified with a confidence of 0.15. (c) By placing cat$_{big}$ next to cat$_{small}$ we observe a significant jump in the classifier's confidence from 0.15 with cat$_{small}$ alone to 0.84 on 2-cats. While (d) SMUG and (e) IG correctly attribute the model confidence to cat$_{big}$, (f) OPTBOX exploits the object rescaling in LSC, favoring the more compact object.

The LSC metric makes a trade-off when optimizing for both compactness and confidence. To analyze this we look at several qualitative examples (Fig. 2) of the bounding boxes identified by the OPTBOX brute-force approach to optimize the LSC metric. We observe that OPTBOX often finds bounding boxes that are very small, typically a sufficiently discriminative region or pattern in the image (e.g., *typewriter keyboard*, *paddlewheel*, *vine snake* in Fig. 2), or the full object if the object is itself small (e.g., *basketball*, *plunger*). In all these cases we find that SMUG highlights several other aspects of the object as well (typewriter's tape; the dog's eyes, nose and ears, etc.)

**OPTBOX can game the LSC metric (Dabkowski & Gal, 2017) to favor compact regions containing redundant features**. We design an experiment where we introduce redundant features and compare the explanations generated by various methods. Fig. 3(c) contains 2 cats - $cat_{big}$ and $cat_{small}$ where $cat_{small}$ is $cat_{big}$ scaled to quarter its size. This image is classified correctly with a confidence of 0.84. If $cat_{small}$ is removed from Fig. 3(c) we get Fig. 3(a) which is classified with a confidence of 0.83 (a marginal drop of 0.01 in the confidence score) and if we remove $cat_{big}$ we get Fig. 3(b) which is classified with a confidence of 0.15 (the confidence drops by 0.68). Thus, $cat_{small}$ is redundant in Fig. 3(c) as its contribution to the prediction is nominal. Hence, the expectation is that a good saliency method would assign greater attribution to $cat_{big}$ than $cat_{small}$. In this respect, both SMUG and IG capture the model's behavior correctly. Additionally, SMUG's attributions lie on the $cat_{big}$'s face while IG's lie on the edges (possibly due to a sharp change in contrast). OPTBOX, however, attributes the model's prediction to $cat_{small}$. This behavior can be explained using Eq. 5. Essentially, OPTBOX employs a brute force search to look for a region as small as possible and classified with a good enough confidence by the model so as to maximize Eq. 5. And by doing so it highlights regions with tightly packed redundant features ($cat_{small}$ in this case). As argued in (Dabkowski & Gal, 2017) LSC does capture certain desirable attributes of saliency maps which makes it a popular choice to evaluate these explanation methods, but in this work we show that saliency methods can be designed to fool this metric.

### 5.3 Text Dataset: Beer Reviews

We present randomly selected qualitative examples from the test set comparing with other methods including SIS, IG, and GROUNDTRUTH in Fig. 4 and Appendix C. The solution of SIS consists of multiple disjoint set of words of varying relevance. A saliency map is constructed by scoring the words in the sets between (0,1] on the basis of relevance of the set. However, evaluation is harder. Unlike images where the masked image can be cropped and resized as input to compute the LSC metric, the same strategy cannot be followed on the text model. Specifically, ImageNet models are trained with extensive data-augmentation including random crops and resizing, and the modified image is less likely to be out-of-distribution. Whereas in the case of text, this model doesn't employ any form of meaningful augmentation, and the masked text is much more likely to come from a distribution that has not been seen during training. Hence, LSC is not applicable here.

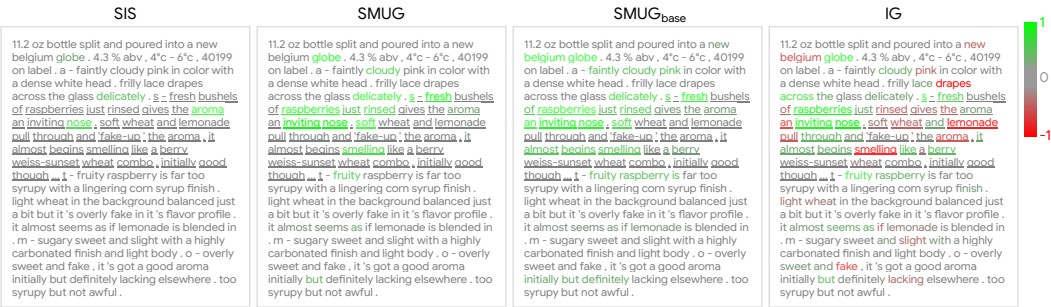

Figure 4: Example comparing our method (SMUG, SMUG$_{base}$) with SIS and IG on a test sample from the Beer Reviews dataset. Green color signifies a positive relevance, red color signifies negative relevance. The underlined words are human annotations. More examples can be found in Appendix C.

## 6 Conclusion

We present an approach that uses SMT solvers for computing minimal input features that are relevant for a neural network prediction. In particular, it uses attribution scores from Integrated Gradients to find a subset of important neurons in the first layer of the network, which allows the SMT encoding of constraints to scale to larger networks for finding minimal input masks. We evaluate our technique to analyze models trained on image and text datasets and show that the saliency maps generated by our approach are competitive or better than existing approaches and produce sparser masks.

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

APPENDIX

Here, we present additional quantitative and qualitative examples supporting the results in Sec. 5. In particular Sec. A shows qualitative examples on MNIST, Sec. B presents quantitative and qualitative comparisons with regard to the choice of key parameters of our proposed SMUG explanation technique. Sec. C presents additional qualitative examples comparing the output of SMUG with other methods on text samples from the Beer Reviews dataset. Sec. D presents several more qualitative examples from ImageNet comparing the saliency masks produced by SMUG with those produced by other methods. In Sec. D.1 we also discuss an example where an explanation technique can be used to identify potential biases of the trained model. Finally, Sec. E presents some examples of concrete constraints that the solver optimizes.

## A  MNIST

In this section, we present some more details about our experiments with MNIST using the full SMT encoding from Eq. 1 in Sec 3.1.

SMT solvers can be used to encode the semantics of a neural network (Katz et al., 2017). In particular, given a fully connected neural network with $n$ hidden layers, weights $W = \{W_1, W_2 \ldots, W_n\}$, biases $B = \{b_1, b_2 \ldots, b_n\}$, activation function $\phi$, and final layer softmax $\sigma$, we can use the SMT theory of nonlinear real arithmetic to obtain a symbolic encoding of the network. Let $X \in \mathbb{R}^{m \times n}$ denote an input image with $m \times n$ pixels, $M \in \{0, 1\}^{m \times n}$ an unknown binary mask, $L_i$ the output (i.e., activations) of the $i^{th}$ layer ($L_0 = X$ is the input) and $\alpha(p_j)$ the output of $j^{th}$ logit in the final layer:

$$L_i \equiv \phi(W_i L_{i-1} + b_i) \qquad \alpha(p_j, W, B, X) \equiv \sigma_j(W_n L_{n-1} + b_n)$$

Given this symbolic encoding, we can encode the minimal input mask generation problem as:

$$\exists M : \mathbf{minimize}(\Sigma_{ij} M_{ij}) \wedge \alpha(p_{label}, W, B, M \odot X) > \alpha(p_l, W, B, M \odot X) \qquad \forall l \neq label \quad (6)$$

where $p_{label}$ and $p_l$ refer to the logits corresponding to the true label and the other labels respectively. The number of constraints grow with increasing network size and SMT decision procedure for Non-linear Real Arithmetic is doubly exponential. Even for piecewise ReLU networks, SMT decision procedures for Linear Real Arithmetic combine simplex-based methods (exponential complexity) with other decision procedures such as Boolean logic (NP-complete complexity), which causes the solving times to grow dramatically with network size. When we apply this encoding even for a small feed-forward network on MNIST dataset, the SMT solver does not scale well (Section 4). This motivates our proposed approach for using gradient information to simplify the SMT constraints.

Table 2 shows the SMT solver runtimes and whether the constraints were solved (SAT). We observe that with a timeout of 60 minutes, the SMT solver could solve the full constraints for only 34 of the 100 images. Another interesting point to observe is that the solver could not solve any of the instances for digits 0 and 3. We also show some of the minimal masks generated by the SMT solver for few MNIST images in Figure 5.

Table 2: **Solver Runtime and SAT instances.** We report the average solver runtime and instances solved per digit with a timeout set at 60 mins.

| Digit | 0 | 1 | 2 | 3 | 4 | 5 | 6 | 7 | 8 | 9 | ALL |
|---|---|---|---|---|---|---|---|---|---|---|---|
| **Runtime (mins)** | N.A. | 31.19 | 45.26 | N.A. | 33.09 | 35.68 | 42.80 | 53.11 | 36.05 | 19.62 | 35.59 |
| **SAT Instances** | 0/8 | 8/14 | 4/8 | 0/11 | 8/14 | 4/7 | 4/10 | 2/15 | 1/2 | 3/11 | 34/100 |

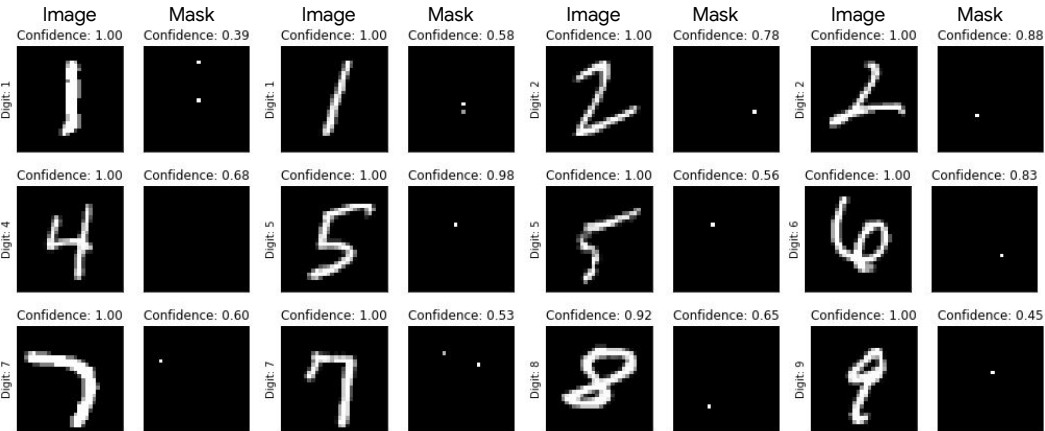

Figure 5: MNIST images and the corresponding masks corresponding to Sec. 3.1.

# B  HYPERPARAMETER CHOICES

In our proposed approach, the choice of top-$k$, and $\gamma$ (Eqns. 3, 4, 5) have an effect on the final quality of the explanations, and the time it takes for the solver to identify the mask. This section presents quantitative and qualitative comparisons for different choices of top-$k$ and $\gamma$.

## B.1  QUANTITATIVE COMPARISON FOR DIFFERENT CHOICES OF TOP-$k$ AND $\gamma$

Fig. 6 presents quantitative comparisons for different choices of top-$k$ and $\gamma$.

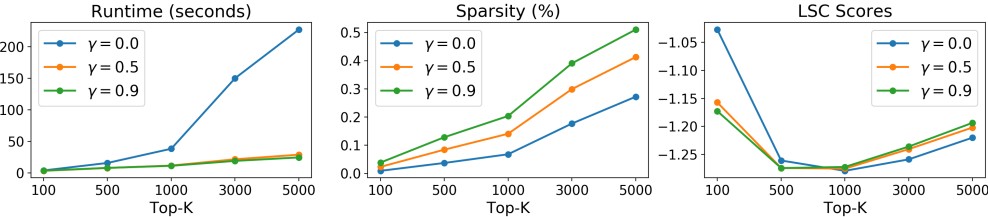

Figure 6:  Hyperparameter Comparision

**top-$k$** We analyze images with top-$k \in \{500, 1000, 3000, 5000\}$. Increasing the $k$ value increases the receptive field and the discovered input masks also grow in size with increasing values of $k$. Figure 6 shows how the solver runtime and the mask size vary with $k$. As expected, larger $k$ values results in larger number of constraints and therefore larger solving times as well as larger mask sizes.

**Gamma** We analyze the effect of $\gamma \in \{0.0, 0.5, 0.9\}$, also shown in in Fig. 6. We observe that by decreasing gamma values, the masks become sparser. The key reason behind this is that with smaller gamma values, the SMTsolver is less constrained to maintain the original neural activations for the selected neurons, and hence can ignore additional input pixels that do not have a large effect. It is also noteworthy to notice that the solver run-time increases with decreasing value of $\gamma$.

## B.2 Top-k vs $\gamma$ on ImageNet - Qualitative examples

Fig. 7 presents qualitative examples of the saliency maps on Imagenet examples for different choices of top-$k$ and $\gamma$.

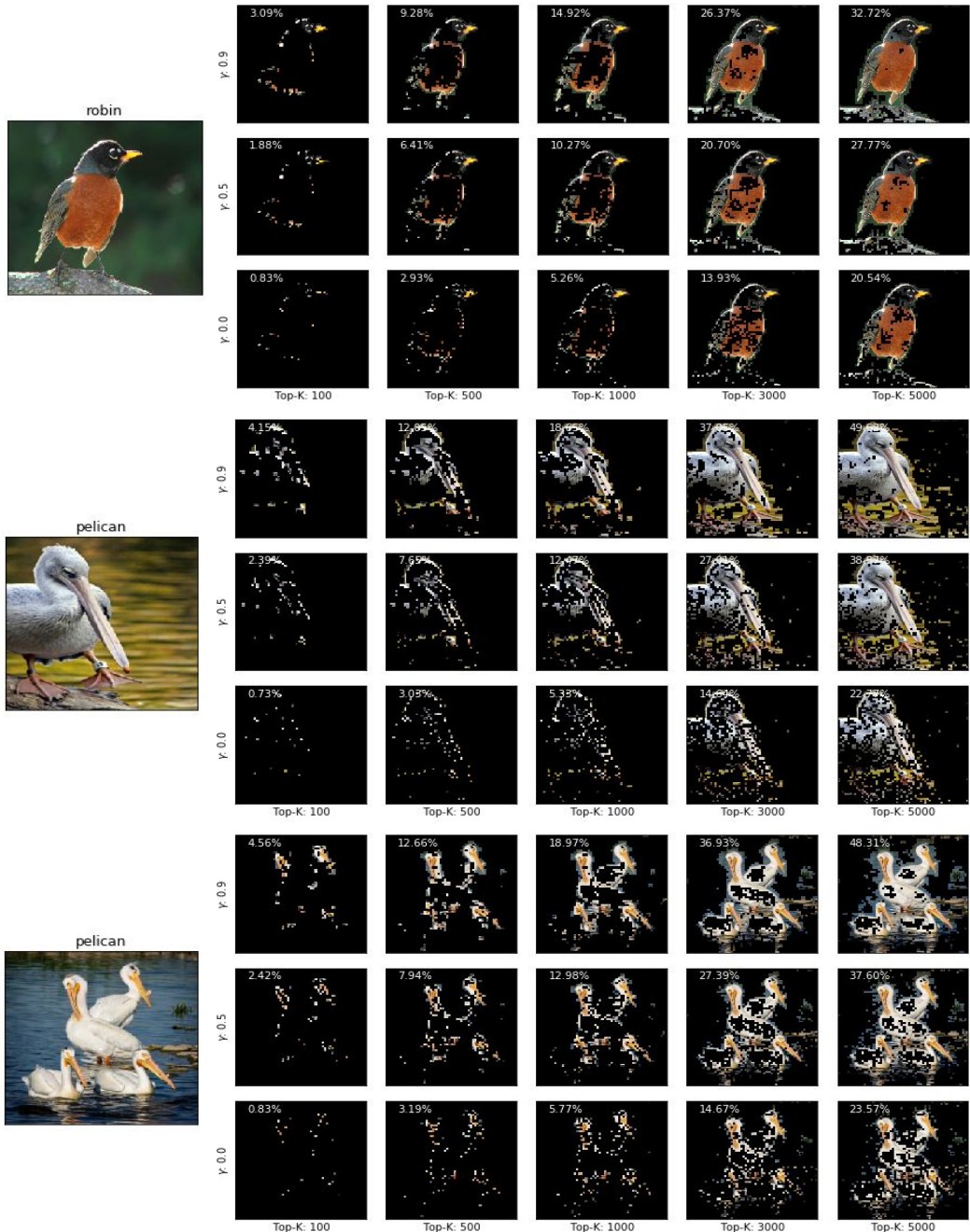

Figure 7: Qualitative examples of the masks generated by SMUG on examples from Imagenet for different choices of top-$k$ (columns) and $\gamma$ (rows). $\gamma = 0$ is the most minimal mask. Even at low values of top-$k$ and $\gamma$ SMUG highlights pixels relevant for the object class.

# C  QUALITATIVE TEXT EXAMPLES

Fig. 8 presents additional examples comparing the output of SMUG with SIS, SMUG$_{base}$, and IG on the Beer Reviews dataset.

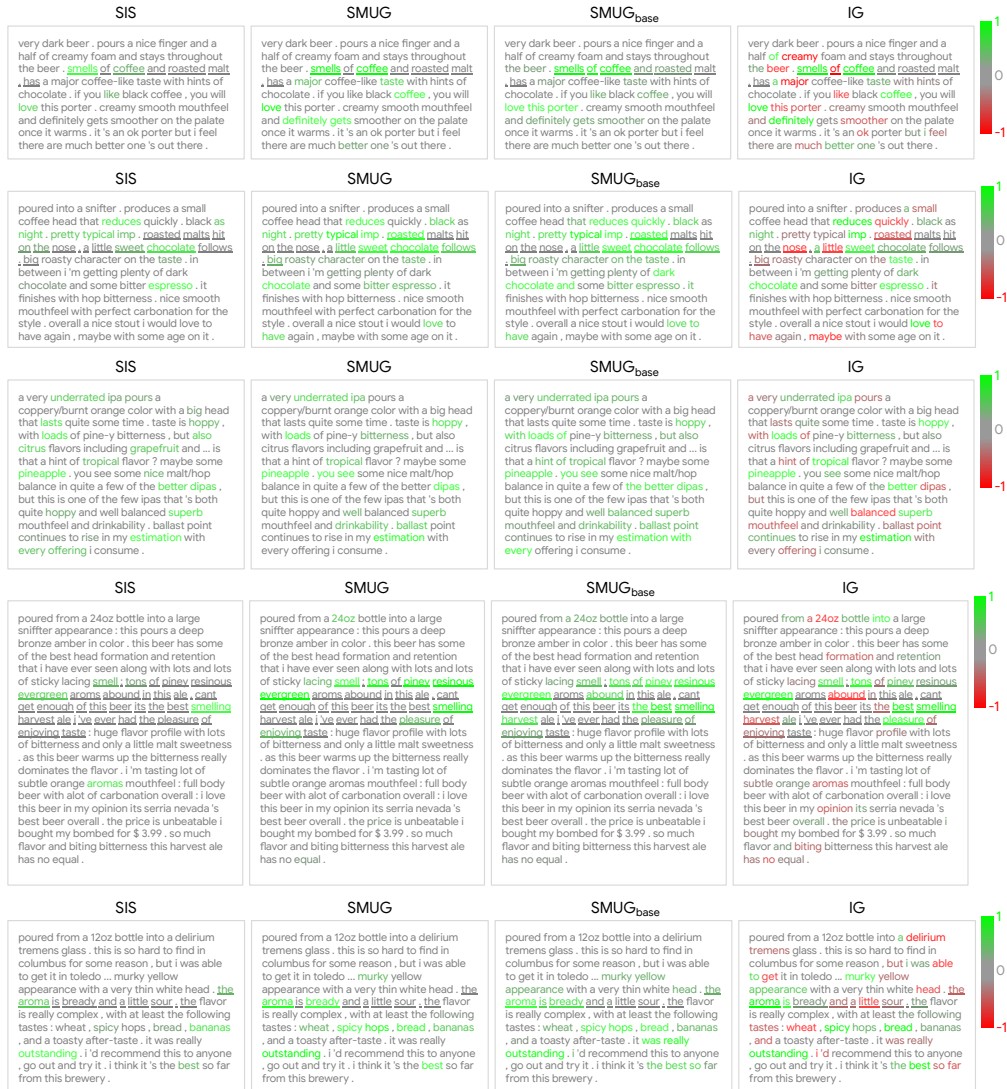

Figure 8: Examples comparing our method (SMUG, SMUG$_{base}$) with SIS and IG on test samples from the Beer Reviews dataset. Green color signifies a positive relevance, red color signifies negative relevance. The underlined words are human annotations.

# D  ADDITIONAL QUALITATIVE IMAGE EXAMPLES: SMUG

Fig. 9 presents boolean masks and the saliency maps produced by SMUG on several ImageNet examples. Fig. 10 and 11 present additional examples comparing the saliency masks and bounding box (for LSC) produced by SMUG, SMUG$_{base}$, and IG.

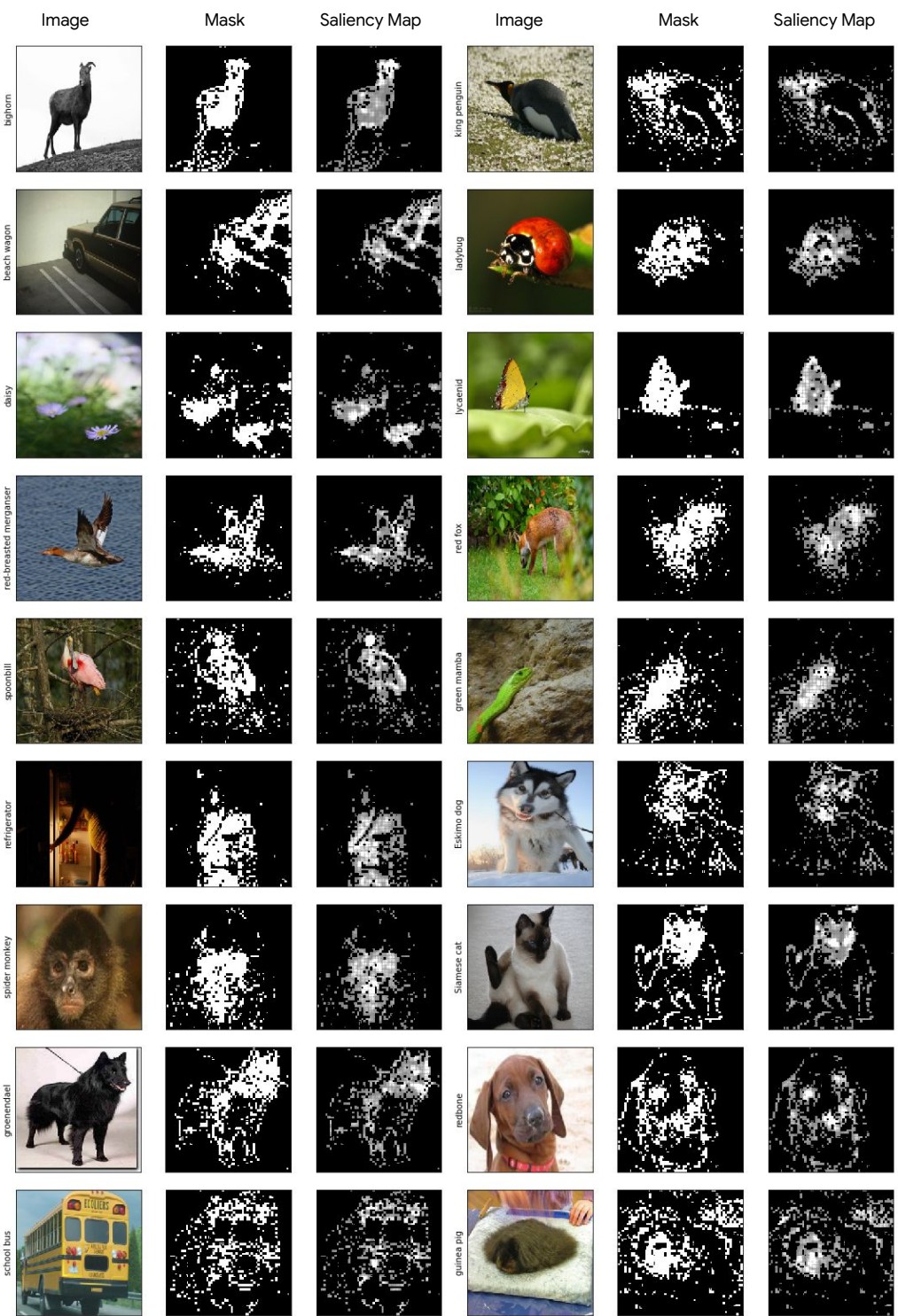

Figure 9: Examples showing the boolean masks and the saliency maps produced by SMUG on several ImageNet examples.

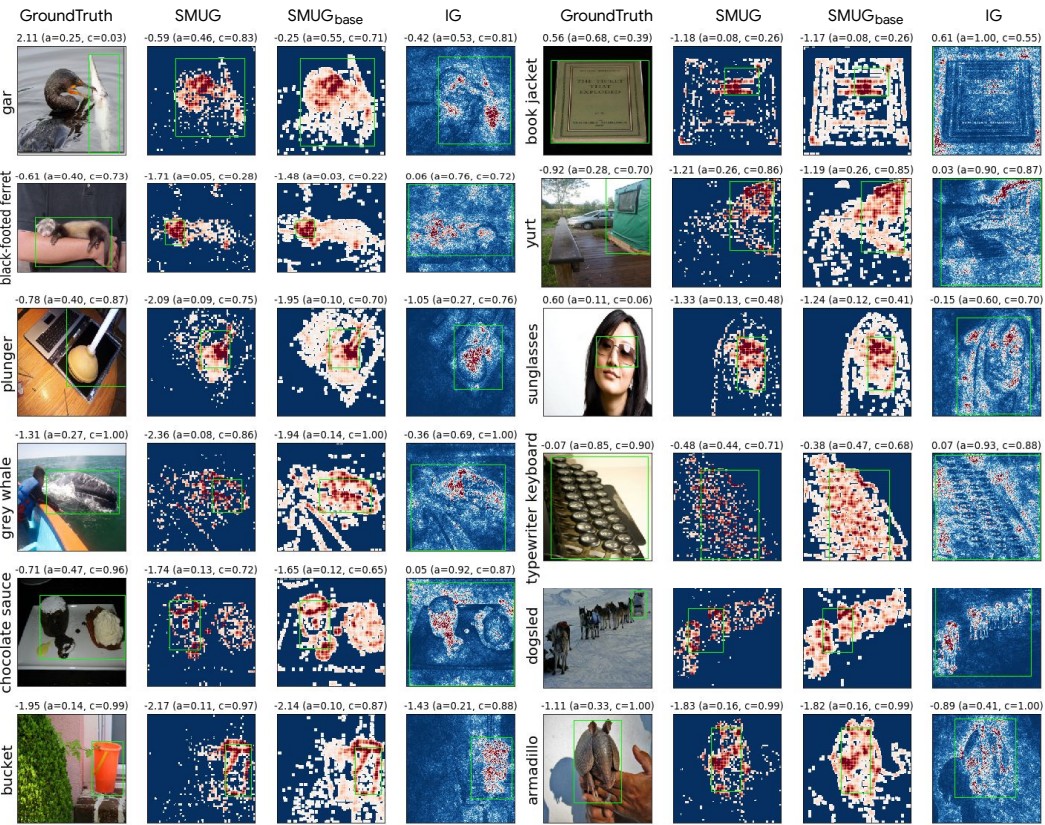

Figure 10: Examples comparing saliency maps where SMUG outperforms SMUG$_{base}$, and IG. The green box on the original image highlights the groundtruth box; for the saliency methods it represents the bounding box with the best LSC score. Numbers on top denote the LSC score, the fractional area of the bounding box ($a$), and the confidence of the classifier ($c$) on the cropped region.

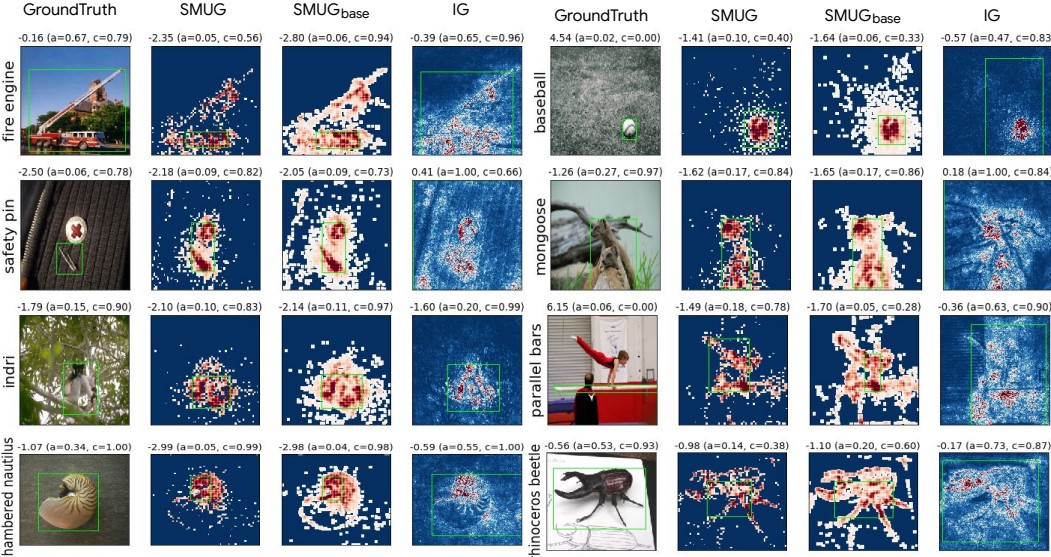

Figure 11: Examples comparing saliency maps where SMUG$_{base}$, or IG outperforms SMUG. The green box on the original image highlights the groundtruth box; for the saliency methods it represents the bounding box with the best LSC score. Numbers on top denote the LSC score, the fractional area of the bounding box ($a$), and the confidence of the classifier ($c$) on the cropped region.

## D.1 Biases

Model explanation techniques can also be particularly useful in studying model biases. Fig. 12 shows some examples where the model correctly predicts the class as "parallel bars" but it appears to actually focus more on the person leaping over the bar to make the prediction as opposed to looking at the bar itself. These can help us understand and identify cases where the model has developed a bias (in this case, based on training data).

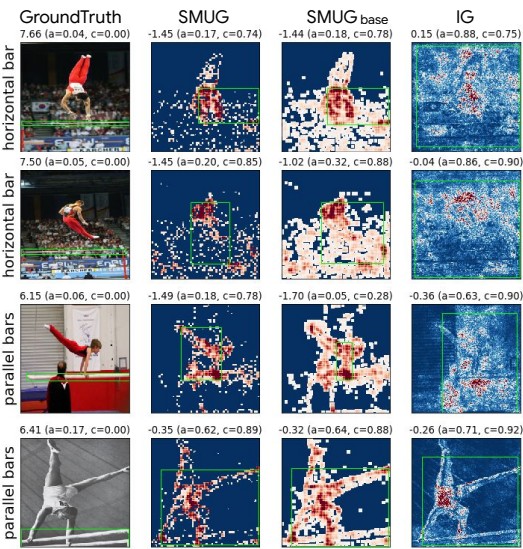

Figure 12: [Model bias] Examples where the model correctly predicts the class as "parallel bars" and "horizontal bars" for the corresponding inputs but the model's focus is on the person leaping over the bar as opposed to the bar (i.e., the predicted object class) itself.

## E SMT CONSTRAINTS

In this section, we present an example set of SMT constraints obtained by our technique for an example image from ImageNet. For brevity, we show the set of constraints for $k = 5$ and $\gamma = 0$. As mentioned in Sec 5.1, each masking variable $M_{ij}$ corresponds to a $4 \times 4$ grid of pixels, where the grid is denoted by $X_{i:i+3,j:j+3}$. For example, the mask variable $M_{132,135}$ corresponds to the pixel grid $X_{132:135,132:135}$. Following Eq. 3, the SMT constraints corresponding for top $k = 5$ IG positive attributions in the first layer are given by:

$$99.53X_{132,132} - 58.37X_{132,136} + 4.88X_{132,140} - 141.25X_{136,132} + 639.97X_{136,136} + 10.29X_{136,140} - 9.66X_{140,132} + 20.30X_{140,136} - 25.19X_{140,140} - 0.58 > 0$$

$$-270.67M_{120,150} + 101.23M_{142,144} + 10.38M_{113,124} + 207.98M_{122,121} + 640.64M_{121,121} - 100.72M_{121,126} + 25.06M_{121,165} - 75.49M_{121,156} + 75.47M_{112,154} - 0.36 > 0$$

$$2925.38X_{144,132} - 395.09X_{144,136} + 81.61X_{148,132} - 999.88X_{148,136} - 82.70X_{152,132} + 17.08X_{152,136} + 0.21 > 0$$

$$-20.87X_{76,80} + 8.40X_{76,84} - 122.72X_{80,80} + 929.71X_{80,84} + 85.52X_{84,80} + 138.99X_{84,84} - 0.01 > 0$$

$$231.34X_{168,148} + 722.71X_{168,152} + 80.18X_{172,148} + 663.96X_{172,152} + 5.37X_{176,148} + 4.63X_{176,152} + 0.12 > 0$$

