# OpenReview forum: "Scaling Symbolic Methods using Gradients for Neural Model Explanation"
_ICLR.cc/2021/Conference — ICLR 2021 Poster_

### Official Review · AnonReviewer3 · 2020-10-28
**Incremental improvement, paper title and pitch is a bit misleading**

**Rating:** 5
**Confidence:** 3

**Review:**

Summary: This paper improves the Integrated Gradient (IG) based model explanations by picking the top-k activated neurons in the first layer and choosing a subset of inputs making sure top-k activated neurons still active enough according to a certain threshold.

I appreciate the authors' contribution to improving IG, but feel certain claims are inappropriate. I hope the authors could help to clarify and correct any misunderstandings I may have.

Although the paper title suggests scaling symbolic methods (e.g. SMT-based approaches), the proposed approach SMUG does not necessarily rely on SMT at all. Since ReLU is not considered by SMUG, integer programming (for encoding whether an input dimension is selected) should be sufficient. The way of scaling symbolic methods appears to be simply "avoid using them". Also, it is unfair to blame that SMT-based approaches for neural network verification are not scalable while SMUG could handle much larger networks, because SMUG really solves a _different_ problem.


Questions:
Q1: The authors argue "From Table-1, we observe that SMUG and SMUGbase achieve a significantly better score (−1.26 and −1.23 resp.) compared to IG (−0.34)."  However, IG actually achieves -0.29, which is more closer to the GroundTruth (-0.34). Does that imply IG is actually better?

Q2: In figure-5, the mask achieves fairly low confidence (e.g. less than 0.7 for most examples). Why is that?
Is it a fundamental weakness even for full SMT encoding?  Furthermore, suppose scalability is not an issue, would SMUG always behave worse than full SMT encoding (thus suffer from the same weakness)?

---

> ### Author Response · Authors · 2020-11-14
> **A different perspective on using gradients with symbolic methods**
>
> Thanks for taking time to review our work. Based on the comments, it appears that we come from somewhat different perspectives. We address the issues, and questions raised below, and hopefully that’ll clarify your concerns.
>
>
> > Incremental Improvement.
>
> While our approach might seem to build on an existing explanation method, we reiterate some of our key contributions:
> * Our method shows a promising direction to combine symbolic methods with gradients. The previous SMT based methods scale to Neural Network with 5000 nodes, for this application, ours scales to Neural Network with millions of nodes.
> * Contribution to masking methods - our approach is more reliable as we handle the OOD examples - masking on linear equations.
> * Solving the issue about choosing a baseline for IG (Strumfels et. al.)
> * LSC score is a popular and widely accepted metric for evaluating saliency maps. We show that one can actually use this score to generate explanations (OPTBOX) and that these explanations aren’t faithful to the model (Section 5.2)
>
>
> > Although the paper title suggests scaling symbolic methods (e.g. SMT-based approaches), the proposed approach SMUG does not necessarily rely on SMT at all
>
> We disagree with this perception. We think a good middle ground to scaling symbolic methods to neural networks lies in combining them with gradient based information to significantly reduce the number of constraints being examined.
>
> > Since ReLU is not considered by SMUG, integer programming (for encoding whether an input dimension is selected) should be sufficient. The way of scaling symbolic methods appears to be simply "avoid using them".
>
> SMT solvers do indeed use Mixed Integer Programming solvers in the backend. Our SMT based approach is more general as it can encode other non-linear constraints and theories in future, as well as allows for more flexible optimization constraints (our current formulation uses minimization constraints). For instance, in our MNIST experiments SMT solvers are needed there as we encode the full neural network with non-linear activations.
>
> > unfair to blame that SMT-based approaches for neural network verification are not scalable ...
>
> In the paper we say SMUG can handle much larger networks than the previous SMT based approaches for model explanations like Property inference for deep neural networks(Gopinath et al., 2019), Abduction-based explanations for machine learning models (ignatiev et al., 2019) and NOT for neural network verification. We agree that it’s a different problem we are solving. . and we will clarify this in the paper.
>
> >  However, IG actually achieves -0.29, which is more closer to the GroundTruth (-0.34). Does that imply IG is actually better?
>
> GroundTruth here basically refers to the bounding boxes which were collected to localize the object in the image and are completely independent of the model. It is very much possible that the model could actually be relying on pixels outside this region for prediction. So these boxes are only a weak proxy for where the model could be looking. In Figure 1(a) we notice that IG distributes high scores more widely (spatially) which causes the size of the bounding box to be large which are almost the size of the ground truth bounding boxes. On average groundtruth masks cover about 50% of the image area and IG’s bounding box covers around 65% of the image area. In contrast, SMUG (mask size 17% of the total image area) and SMUG_base (mask size 43% of the total image area) tend to assign high scores to a much more localized set of pixels which causes the LSC score to improve.
>
> > In figure-5, the mask achieves fairly low confidence (e.g. less than 0.7 for most examples). Why is that? Is it a fundamental weakness even for full SMT encoding? …
>
> To clarify, confidence is the softmax score of the classifier for that image, label pair. Confidence scores of 0.6 and 0.7 for the masked inputs aren’t infact low given that the masked image contains ~ 1 or 2 pixels out of 728  pixels (each white dot in Figure 5. represents 1 pixel) and yet is classified correctly with such a high confidence! This tells us that encoding the full neural network to produce explanations often finds adversarial artifacts which just happen to be classified correctly. Thus, finding minimal regions which are classified correctly is a bad way of generating explanations something which the full SMT encoding and the existing masking methods like LIME(Alvarez-Melis & Jaakkola, 2018), RISE (petsuik et. al.), Interpretable Explanations of Black Boxes by Meaningful Perturbation (Fong et al.) do. So, SMUG would always produce more reliable explanations  than these  methods.

---

### Official Review · AnonReviewer1 · 2020-10-28
**Interesting idea about using IG for fast mask generation**

**Rating:** 7
**Confidence:** 4

**Review:**

This paper provides an interesting pos-hoc explanation method to identify relevant features in an input that may inform a trained neural model's prediction. The task is to identify a binary mask over input image/text such that the masked input yields almost similar prediction as original input. The author formulates this as an SMT solver task, but instead of making sure that the output prediction is similar (which involve multiple time consuming pass over potentially huge networks), they make sure that high influential neurons in first layer of the network are still activated. This provides a less time consuming way to evaluate invariance of masked input.

The evaluation is done on multiple datasets - ImageNet, MNIST and Beer Reviews. The authors compare against previous continuous saliency baselines (IG and GradCAM) and full SMT solver method SIS, in addition to ground truth annotations for bounding boxes. They find that their method, called SMUG, is better performing in terms of LSC metric (which measure the log ratio of percentage of pixels preserved vs confidence in ground truth prediction). They also do some qualitative analysis and found that SMUG tend to produce low area masks. They do not perform any systematic human study to evaluate their method.

The paper as written is quite clear and I was able to parse the information with reasonable ease.

As it is, I will recommend acceptance for the paper. But I believe addressing following comments will make this paper better.

Questions/Comments:
1. The LSC metric is combines the two quantities that don't have any direct scaling relation with each other - area of box and confidence in prediction. For example, going from 0.5 to 0.6 is more important than going from 0.8 to 0.9 in terms of confidence. The same cannot be said for the area. Putting the two together hides information necessary for comparison. For example, are the low values for IG mainly because of large bounding boxes ? I would suggest presenting both results separately along with LSC.

2. A proxy for human study would be to measure the overlap of SMUG results with provided bounding boxes.

3. Why does bilinear interpolation not move data out of distribution ? For example, focusing on face of cat as bounding box and then blowing up should clearly put it OOD ? Am I missing something here ?

4. I am not clear on why restricting your analysis to top-k influential neurons is correct. What if the masked image activate the non-influential neurons -- one see no reason why it won't change the prediction. The IG values are only true for the original image. For example, say for original image, the set of influential neurons is IF and non-influential ones are NIF. If the SMUG masked image activates all of IF neurons and only 10% of NIF neurons, we can get completely predictions. And vice versa. I would suggest running an experiment (for even a small task) where all neuron values are considered and see what the behavior is. (If you have done this experiment, please point me to the relevant section in the paper -- I can't seem to find it).

5. While they are not pos-hoc methods, following paper also try to learn masks as form of interpretation. Please consider citing them -
https://arxiv.org/pdf/1802.07814.pdf , https://arxiv.org/abs/1606.04155, https://www.aclweb.org/anthology/P19-1284v1.pdf, https://arxiv.org/abs/2005.00115, https://arxiv.org/abs/2004.14992

---

> ### Author Response · Authors · 2020-11-14
> **Identifying and retaining influential neurons using gradients helps reducing the SMT constraints and enables scaling**
>
> Questions raised:
> Thanks for your positive rating, comments, and questions. We have provided clarifications with regard to your questions below.
>
> > they make sure that high influential neurons in first layer of the network are still activated. This provides a less time consuming way to evaluate invariance of masked input.
>
> We wish to clarify that SMUG isn't faster than IG, but it does resolve some key issues compared to IG. Our method would infact be faster when compared to other methods like SmoothGrad (Smilkov et al., 2017), or IG with multiple baselines etc. (Strumfels et al.) which also try to address the same issues.
>
> > They do not perform any systematic human study to evaluate their method.
>
> We found it hard to formulate an appropriate crowdsourcing experiment to help us better evaluate our results, since the main goal of this paper is to identify features important for a model's predictions. While it could be interesting to evaluate similarity between the features considered important by a “model”, and the features identified by “humans, but we believe this is orthogonal to the problem we are solving.
>
> > Q1 The LSC metric is combines the two quantities that don't have any direct scaling relation with each other .. I would suggest presenting both results separately along with LSC
>
> LSC metric was proposed in Real time image saliency for black box classifiers (Dabkowski & Gal, 2017,NeurIPS ‘17) and  is widely used.  Exact numbers for area and confidence used to compute the LSC score has been presented in the title of each image in Figure 1. As per your suggestion, we have now also included sparsity scores in our results in Table 1.
>
> > Q2 A proxy for human study would be to measure the overlap of SMUG results with provided bounding boxes.
>
> Comparing similarities between the explanations generated by our methods and the provided bounding boxes may not be the best option as we are trying to interpret what the model has learnt, instead of what a human might think is most relevant for a prediction.
>
> > Q3 Why does bilinear interpolation not move data out of distribution
>
> Bilinear Interpolation doesn’t solve the problem of OOD. As the Inception model was trained on image crops (which were scaled up), a fair number of the cropped and resized images would be in distribution (since imagenet models use augmentations using random crops).
>
> > Q4 I am not clear on why restricting your analysis to top-k influential neurons is correct. What if the masked image activate the non-influential neurons (NIF)
>
> To reiterate, our goal isn’t to find a mask for which the masked image is classified correctly. Directly masking an image would make it OOD and misclassify it. Such an approach doesn’t produce reliable explanations as we show in the MNIST experiment where adversarial masks are learnt which happen to be classified correctly. By restricting our analysis to top-k nodes our goal is to identify the pixels that are contributing to the information captured by those nodes. IG attributions on the the first hidden layer inform us of the nodes that are important. And as the nodes capture important information, the corresponding pixels are important too. As we don’t do a forward pass of the masked image. We are not concerned  whether the masked image activates the NIF neurons as long as the IF neurons are active. Hope this clarifies your concern.
>
>
> > Q4 experiment (for even a small task) where all neuron values are considered..
>
> Considering all neurons is part of what makes scaling SMT hard. Which is why looking at top-k neurons is a key component of our approach as described earlier.
>
> Thanks also for pointing us to additional references, we will describe their relationship with our work in the related work section.

---

### Official Review · AnonReviewer2 · 2020-10-30
**Interesting work, but the need for it is unclear**

**Rating:** 5
**Confidence:** 3

**Review:**

This paper addresses the question of identifying which input features are most important for a neural network's decision. To do so, it frames the problem as an SMT problem that seeks to select the best input features, without changing the state of the first layer too much. The paper shows experiments, primarily on image classification, and also examples of how the approach may be applied to text classification.

There are a couple of things that the paper combines in its final work: (a) using integrated gradients to score the first layer nodes, (b) restricting the search to only the top-k layer one neurons, and (c) applying SMT to find the subset among them that best preserves the layer 1 activations. The experiments do a good job of comparing the contribution of these components.

However, from the experiments, it looks like the third step (i.e. SMT) is not really necessary, both from the quantitative and qualitative results. The paper argues in the paragraph titled SMUG vs SMUG_{base} that the latter does not provide sparse input features. It is not clear why sparsity is a desirable quality here. We are looking for image regions that best contribute to the output, in which case smoothness in the regions may even be preferred over sparse explanations that pick out specific pixels. It would be good to clarify this.

The paper claims that it is not a good idea to use an SMT based explanation that focuses on the final layer. The complexity argument makes sense but could perhaps be mitigated by an extension of the top-k strategy, perhaps. But the argument that the masked inputs are out of training distribution is not mitigated by the proposed approach. If the masked inputs are out of distribution for the full network, then certainly they are out of distribution for the layer one activations too (eq 2). Is the out-of-distribution issue really a problem, and if so, why doesn't it affect the first layer?

The SMT problems in (2) and (3) ask for a mask such that for every node in the k selected ones, the first layer activation with masked inputs is more than a scaled version of the first layer activation for the original input. This one-sided inequality seems to be tied to the fact that the experiments use ReLU activations. For example, if an activation for a certain neuron is highly negative, and we use a different activation function (say tanh), then we would want a similarly negative output for the masked inputs too. It may be worth mentioning this somewhere (maybe a footnote).

The experiments on images seem interesting (with the caveat about the need for SMT at all). But the experiments with text only show examples, which makes it only anecdotal. It would be interesting to compare to the results of crowd sourcing experiments where the most important words are chosen, or perhaps asking turkers to pick from the words selected by the proposed method and other interpretability methods (e.g. HotFlip, etc).

---

> ### Author Response · Authors · 2020-11-14
> **Sparsity can be desirable in other applications.**
>
> Thanks for your comments and bringing up some key issues. We address them below, and hopefully that also helps clarify the need for our approach.
>
> > need for it is unclear
>
> Our perspective is that symbolic methods would benefit from the use of gradients with one application being explanation. Our method shows a promising direction to combine symbolic methods with gradients. In our case, with the addition of the third step we think it is in fact useful to get more compact and sparse regions for the explanation (more below).
>
> >It is not clear why sparsity is a desirable quality here. .... smoothness in the regions may even be preferred over sparse explanations that pick out specific pixels. It would be good to clarify this.
>
> Smoothness in regions might be preferable in some applications such as localizing salient objects in images. However, there are applications such as in medical contexts where lesions / pathologies are sparser and less coherent where incorporating smoothness in the region might be less desirable. The same may be said of finance or text applications. Sparsity is desirable as only the most important features are kept. Inducing sparsity removes False Positives from the mask found by SMUG_base (more in the below comments).
>
> >However, from the experiments, it looks like the third step (i.e. SMT) is not really necessary, both from the quantitative and qualitative results.
>
> Our quantitative and qualitative evaluations do show the advantage of SMT, and its effect on sparsity. In particular, we have now added sparsity scores for each method to Table 1 (previously in Sec 5.1). As mentioned in the paper, LSC scores use the compactness of the bounding boxes as a proxy for the sparsity of the saliency map. As we can see in Figure 1(a) and 1(b) that the bounding boxes for SMUG and SMUG_base have similar sizes even though the former produces much sparser saliency maps. This causes both SMUG and SMUG_base to receive similar LSC scores. Thus, the LSC scores alone don’t reveal the full picture and we need to consider the sparsity scores as well. On average, SMUG_base’s mask size is 43% and SMUG’s mask size  is 17% of the image area. SMUG (SMT) removes **67%** of the pixels from SMUG_base while keeping the LSC score approximately the same (there’s a slight improvement). Again, the improvement wasn’t dramatic only because LSC score fails to account for the sparsity of the saliency map. Furthermore, as the removed pixels didn’t cause the LSC scores to drop we infer that the relevant pixels  were actually retained.  Qualitative: In Figure 1(b) we see that even for the examples where SMUG and SMUG_base receive equal scores, SMUG removes a lot of less relevant pixels.

---

> ### Author Response · Authors · 2020-11-14
> **OOD issue is mitigated with SMUG**
>
> >  But the argument that the masked inputs are out of training distribution is not mitigated by the proposed approach. If the masked inputs are out of distribution for the full network, then certainly they are out of distribution for the layer one activations too (eq 2). Is the out-of-distribution issue really a problem
>
> This is a very interesting point, and it is in fact mitigated by our method. The reason why out-of-distribution inputs are relevant is because several methods - LIME(Alvarez-Melis & Jaakkola, 2018), RISE (petsuik et. al.), Interpretable Explanations of Black Boxes by Meaningful Perturbation (Fong et al.) optimize their mask by doing a forward pass on perturbed images that are potentially OOD. However, using SMT to mask inputs in the first layer can be thought of as analogous to examining co-efficients for a linear regression model (we deal with linear constraints). We do not do a forward pass on the perturbed/masked images (potentially OOD) unlike the other methods; instead, we rely on model gradients on the original input image alone. So, in our case we do indeed mitigate this issue.
>
> > complexity argument makes sense but could perhaps be mitigated by an extension of the top-k strategy, perhaps
>
> Perhaps. For large models like ImageNet, solving for the encoding of the second layer neurons causes  the SMT solver  to time out (the timeout was set at 24 hours). One other option is to run experiments on the MNIST dataset but the issue there is assessing the quality of the saliency maps. LSC metric can’t be used here as the model isn’t trained on image crops like Inception, hence computing the LSC score using image crops would make the input out of distribution. Hence, it would be hard to tell if there’s an improvement in the quality of the saliency maps without a metric or visually.
>
>  > This one-sided inequality seems to be tied to the fact that the experiments use ReLU activations. … Worth mentioning
>
> Thanks, we have updated our submission with this note (footnote 1)
>
>
> >  the experiments with text only show examples, which makes it only anecdotal. It would be interesting to compare to the results of crowd sourcing experiments
>
> It’s unclear how crowd sourcing experiments would help us better evaluate our results, since the main goal of this paper is to identify features important for a model predictions. While it could be interesting to evaluate similarity between the features considered important by a “model”, and the features identified by “humans, but we believe this is orthogonal to the problem we are solving.

---

### Official Review · AnonReviewer4 · 2020-11-05
**Interesting and noval (AFAICT) combination of integrated gradients and SMT that leverages strengths of each.  Some choices need more justification and explanation.**

**Rating:** 7
**Confidence:** 3

**Review:**

## Summary

This paper presents a method to encode the minimal input feature discovery problem -- finding the minimal set of features in a input that is necessary for a prediction -- into a form that can is amenable to satisfiability modulo theory (SMT) solvers.  In particular they first use the integrated gradients methods to score first-layer neurons on the degree to which they influence the prediction.  Then, they produce and solve an SMT problem that finds the minimal mask that changes these influential neurons.  They demonstrate their approach on several problems.

## Review

Overall I thought this was an interesting paper with practical utility.

- The formulation is interested and is a novel balance of quite different methodologies with useful results
- The paper is clear and fairly well written, but some higher level intuition about the approach would help
- I'd like to see some more justification for focus on the first layer, and experiments (described below)

In section 3.2 you mention that you use IG to "score the neurons in order of relevance by treating the first layer activations as an input to the subsequent network".    It's not clear to me whether the $d$ in the function $F: R^{a\times b \times c} \to [0, 1]^d$ is the set of all nodes in the neural network or just the first layer or just the final layer?   It seems the latter is the case, based on Equation 2.

More generally, it's not clear to me what privileges the first layer in this work (Eq 2).  My understanding is that
1. simply that restricting attention to the first layer allows SMT to be applicable
2. You use IG to integrates information from all layers, and by restricting Eq 2 to $D^k$ you are effectively combining both methods

This leads to an experimental question: do your explanations improve if you include more than one layer?  This seems like something that is easily testable, at least on small examples.

Writing wise, some of the terms could me more clearly defined.  For instance in the definition of $F$ above, I am left guessing as to what what $a$, $b$, $c$ and $d$ are, and assume they are simply place holders.  Similarly, sometimes we have $N_\theta(x)$ and sometimes $N_\theta(X)$.

---

> ### Author Response · Authors · 2020-11-14
> **First layer is an important focus and we believe combining gradients with SMT here is a first step towards scaling**
>
> Thanks for your positive notes, comments and suggestions. We address your questions and provide clarifications below.
>
> > More generally, it's not clear to me what privileges the first layer in this work (Eq 2).
>
> Your understanding is correct. The main reason to use the first layer is for two reasons: 1) to eliminate the need of doing a forward pass of the masked inputs which are potentially out of distribution, 2) scalability of SMT encoding. Focusing on important neurons deemed important by IG that integrates information from all subsequent layers helps solve both of these challenges.
>
> > This leads to an experimental question: do your explanations improve if you include more than one layer? This seems like something that is easily testable, at least on small examples.
>
> For Imagenet, encoding neurons in the second layer causes the SMT solver to timeout (the solver timeout was set at 24 hours). One other option is to run experiments on the MNIST dataset but the issue there is assessing the quality of the saliency maps. LSC metric can’t be used here as the Neural Network isn’t trained on image crops unlike inception, and while cropping and resizing parts of the image to compute the LSC score, the input would be out of distribution. Hence, it would be hard to tell if there’s an improvement in the quality of the saliency maps without a metric / visually.
>
> > It's not clear to me whether the function  $F:R^{a \times b \times c} \rightarrow [0,1]^d$  is the set of all nodes in the neural network or just the first layer or just the final layer? It seems the latter is the case, based on Equation 2.
>
> This F in IG often refers to the full network (from the input to the final softmax). In our case, it refers to the nodes in the network’s second layer through the final layer. a, b, and c refer to the dimensions of the tensor input to the network (analogous to width, height, and channels) and d is the number of output nodes in the final softmax layer (number of classes). We apologize for the confusion, and have clarified this in the paper. We’ve also fixed the inconsistencies with the notation $N_{\theta}(X)$ in the main paper.

---

### Author Response · Authors · 2020-11-24
**Thanks for the suggestions and comments. Overall Response**

We thank all reviewers for their thoughtful and helpful comments. We have responded to each reviewer and clarified specific issues. We reiterate some of our key contributions:
* Our method shows a promising direction to combine symbolic methods with gradients. While previous SMT based methods for model explanation (Gopinath et al., 2019, Ignatiev et al., 2019) scale to Neural Network with 5000 nodes, we demonstrate our approach on much larger and more complex image and text models.
* Contribution to masking methods:  Our approach which does masking on linear equations is more reliable than other masking methods as it overcomes / avoids the issue of handling OOD samples. (more in our response to AnonReviewer2).
* Further, our solution resolves the issue about choosing a baseline for IG (Strumfels et. al.)
* Another interesting contribution we make is with regard to our understanding of the popular and widely used LSC score metric for evaluating saliency maps. We show that one can actually game this score to heuristically generate explanations (OPTBOX) that  aren’t necessarily faithful to the model (Section 5.2)

Based on the current reviewer suggestions and feedback, we have updated our current submission (with changes highlighted in blue). We'll be happy to incorporate any further suggestions or clarify any other concerns requested by the reviewers.

---

### Decision · Program_Chairs · 2021-01-07
**Final Decision**

**Decision:**

Accept (Poster)

**Comment:**

This paper combines considers the task of finding a minimal set of inputs that explain predictions of trained neural models. The authors propose a method that they refer to as "scaling symbolic methods using gradients" (SMUG). This method use integrated gradients methods to score first-layer neurons on the degree to which they influence the prediction and then produces and solve an SMT problem (restricted to first-layer activations) that finds the minimal mask that changes these influential neurons.

Reviewers had somewhat mixed perspectives on this submission. All reviewers were broadly in agreement that the paper is clearly written and presents an interesting combination of symbolic (i.e. SMT-based) and gradient-based methods for model explanation. R2 questions the need for sparsity (and therefore the SMT component) in model explanations, and R3 similarly notes that SMUG does not necessarily rely on SMT at all. That said, no reviewers raise major concerns with the quality of exposition, experimental evaluation, or the level of technical contributions in this work. The metareviewer is inclined to say that this work is above the bar for acceptance, and represents a reasonable approach to integrating SMT-based and gradient-based methods for model explanation.